# Nutritional Composition, Efficacy, and Processing of *Vigna angularis* (Adzuki Bean) for the Human Diet: An Overview

**DOI:** 10.3390/molecules27186079

**Published:** 2022-09-17

**Authors:** Yao Wang, Xinmiao Yao, Huifang Shen, Rui Zhao, Zhebin Li, Xinting Shen, Fei Wang, Kaixin Chen, Ye Zhou, Bo Li, Xianzhe Zheng, Shuwen Lu

**Affiliations:** 1Food Processing Research Institute, Heilongjiang Academy of Agricultural Sciences, Harbin 150086, China; 2Heilongjiang Province Key Laboratory of Food Processing, Harbin 150086, China; 3China School of Engineering, Northeast Agricultural University, Harbin 150030, China

**Keywords:** adzuki bean, polysaccharides, polyphenols, health benefit

## Abstract

Adzuki beans are grown in several countries around the world and are widely popular in Asia, where they are often prepared in various food forms. Adzuki beans are rich in starch, and their proteins contain a balanced variety of amino acids with high lysine content, making up for the lack of protein content of cereals in the daily diet. Therefore, the research on adzuki beans and the development of their products have broad prospects for development. The starch, protein, fat, polysaccharide, and polyphenol contents and compositions of adzuki beans vary greatly among different varieties. The processing characteristic components of adzuki beans, such as starch, isolated protein, and heated flavor, are reported with a view to further promote the processing and development of adzuki bean foods. In addition to favorable edibility, the human health benefits of adzuki beans include antioxidant, antibacterial, and anti-inflammatory properties. Furtherly, adzuki beans and extracts have positive effects on the prevention and treatment of diseases, including diabetes, diabetes-induced kidney disease or kidney damage, obesity, and high-fat-induced cognitive decline. This also makes a case for the dual use of adzuki beans for food and medicine and contributes to the promotion of adzuki beans as a healthy, edible legume.

## 1. Introduction

Legumes are considered a beneficial part of traditional plant-based diets around the world and are recommended in dietary guidelines from several organizations due to their nutritional status and wide range of potential health benefits [1]. *Vigna angularis* (Willd.) Ohwi et Ohashi is a kind of traditional legume food, also known as red beans or adzuki beans, which is native to China (Figure 1). Adzuki beans are grown all over the world, including Africa, Europe, and the Americas, with Asia having the largest area of adzuki beans. There are about 24 countries in the world planting adzuki beans; in addition to China with the largest production area, Japan, North Korea, South Korea, Australia, Thailand, India, Myanmar, the United States, Canada, Brazil, Colombia, New Zealand, Zaire, Angola, and the former Soviet Union in the Far East all have a certain production area [2]. At present, both the cultivation area and the total production of adzuki beans in China are in first place in the world, and they are also exported to Japan, Malaysia, Korea, the Philippines, and other countries [3]. Adzuki beans belong taxonomically to *Vigna Savi* in the Leguminosae family and are annual herbs that are frequently used as both medicine and food because of their nutritional value and the presence of active substances [4,5]. It is a self-pollinated, diploid plant, and the genome was mapped in 2015, which is of great importance for molecular breeding, as well as genetic studies of adzuki beans [6].

Adzuki beans have high receptivity due to their elegant flavor and enjoyable taste, with rich nutrition and richness in starch and protein. As a feasible source of carbohydrates, the starch of adzuki beans contains a large amount of resistant starch, which does not pose a significant burden on blood glucose after a meal [8,9,10]. Adzuki beans also contain a lot of dietary fiber and rich trace elements for body health, such as iron, calcium, and phosphorus; the contents of phosphorus, manganese, sulfur, cobalt, cuprum, and nickel in the seed coats of adzuki beans are higher than those of mung beans and black beans [5,11]. For Asians, adzuki beans are a great ingredient that can be prepared in a variety of foods (e.g., paste in pastries, desserts, cake, porridge, adzuki rice, jelly, adzuki milk, and ice cream) for at least one billion people [12]. Over the years, the products of adzuki beans have been developed, mixing bean flour with cereal flour to make bread, pastry, and nutritious food for people, as well as extracting isolated protein, starch, and fiber for applications in baby food, soft drinks, pudding macaroni; adding high amounts of dietary fiber into baked goods by using bean-coat flour; applying new extraction techniques to extract bioactive ingredients, such as choline, phospholipids, and so on; and developing bean convenience foods, such as canned beans, candied beans, and edible bean sprouts [13]. Balanced dietary structure plays an important role in improving peoples’ health levels. Therefore, research on adzuki beans and the development of related products help to improve dietary nutrition and balance dietary structure. Although several species of the Vigna sp. have been described in various reviews in recent years [14,15,16,17,18,19,20,21,22,23,24,25,26,27,28], few reviews have been conducted for adzuki beans (Appendix A). Therefore, we want to provide a comprehensive evaluation of the nutritional composition, processing characteristics, and human health benefits in order to provide some assistance in the research and development of adzuki beans.

## 2. Nutritional Composition

### 2.1. Starch

Microscopic observations display that adzuki bean starch is round or oval, with few fissures on the surface, a complete outer edge, a granule diameter length greater than its width, an obvious polarized cross, an umbilical point located in the center of the starch granule, and the existence of a whorl structure similar to the annual rings of trees. The starch of adzuki beans has a low pasting temperature, high paste viscosity, poor cold and heat stability, and easy aging characteristics [29]. The starch particle sizes of different varieties of adzuki beans are between about 34.39~43.29 μm [30,31], even as high as 57.91~83.32 μm [32]. These differences are due to the varieties of adzuki beans or starch extraction methods.

There are some differences in total starch content among azuki bean varieties. It was found that, in early studies, the starch contents of 10 Chinese adzuki bean varieties ranged from 28.50% to 42.09%, and the amylopectin contents were 24.37% to 37.30% of the total seed [32]. Later studies showed that the total starch content of azuki beans could range from 44.55% to 53.92%, including 11.08~26.19% amylose [8]. In 2019, the starch contents of six adzuki bean species determined by Feng et al. ranged from 53.92% to 60.69%, with amylopectin accounting for 66.94% to 70.14% of the total starch content [30]. Another study also determined that the amylose content was 28.30~35.43% of the total starch content in adzuki beans [9]. The total starch content of adzuki beans can reach from 28.50% to 60.69%, among which the amylose and amylopectin contents vary greatly, with the amylose content from 11.08–75.63% and amylopectin content from 24.37% to 88.92%. These differences may arise from the varieties of adzuki beans, origins, years, and periods of seed collection of adzuki beans.

### 2.2. Protein

The average protein level of azuki beans is 16.33~29.2%, which is 2–3 times higher than those of cereal grains. The amino acid composition of azuki bean protein is in a balanced level, and they all meet the Food and Agriculture Organization of the United Nations (FAO)/World Health Organization (WHO) label [33,34]. The protein contents of 10 Chinese adzuki bean varieties measured in a 2011 study ranged from 18.34 g/100 g to 23.81 g/100 g [32]. A protein content analysis of 17 Chinese adzuki bean varieties revealed that the protein mass fractions of azuki beans ranged from 22.83% to 25.41% [8]. An amino acid analysis for different varieties of azuki beans revealed that all the varieties contained eight essential amino acids for humans, with high levels of lysine, leucine, phenylalanine, and valine, with lysine as high as 1.98~1.82 g/100 g. These amino acids are deficient in cereal protein, which reflects the unique nutritional value of the beans [35].

In a detailed study of adzuki bean protein, it was found that about 50% of the protein was globulin, with the highest as 7S globulin, followed by 11S globulin. A circular dichroic analysis showed that the protein was rich in α -helix and β -angle [36,37], where the thermal polymerization caused the formation of disulfide bonds. The improvement of its nutritive and digestibility undergoing heating treatment depended on the composition of amino acids and thermal aggregation in the globulin [38].

### 2.3. Fat

The fat content in adzuki beans is at a low level, but there was a distinct difference in the fatty acid contents of adzuki bean varieties, some at 0.4~1.3 mg/g in adzuki beans [9] or 0.34~0.72 mg/g [31], as well as at 0.48–0.56 mg/g in 2021 [39]. The total fat contents of adzuki beans can range from 0.34 to 1.3 mg/g. There is also experimental confirmation of fatty acid content at about 5.45~6.31 mg/g, of which the dominant fatty acid, palmitic acid, was 24.03~29.41%, linoleic acid content was 30.11~36.12%, and linolenic acid was 23.52~27.76%. These fatty acids accounted for 87.4% of the total fat. In addition, the fatty acid content of stearic acid was 5.46~9.40%, and the oleic acid content was 4.51~7.01% [8]. These differences in the results of studies on fat content determination mainly may be due to differences in adzuki bean varieties.

### 2.4. Polysaccharides

Legumes are rich in protein, fiber, various micronutrients, and polysaccharides [40]. The main active components of legumes are polysaccharides, which have antioxidant and immunomodulatory activities [41,42,43]. Polysaccharides, as long-chain macromolecular compounds, consist of more than 20 monosaccharides condensed by glycosidic bonds, which can be broken down into monosaccharides and oligosaccharides by the action of enzymes or acids [44]. The biological activity of polysaccharides depends mainly on the composition of the sugar, its molecular weight, and the type of glycosidic bond [41]. In 2015, water-extractable and alkali-extractable polysaccharides were obtained from the seeds of adzuki beans. The water-extractable polysaccharides were composed of rhamnose, arabinose, mannose, galactose, and glucose. The alkali-extractable polysaccharides were composed of rhamnose, arabinose, mannose, galactose, and galacturonic acid [42]. In recent research, it was found that the polysaccharides in azuki beans were mainly composed of three fractions, with molecular weights of 131 kDa, 83 kDa, and 5 kDa, respectively. Arabinose, galactose, glucose, xylose, mannose, and galacturonic acid in the polysaccharides of azuki beans were found to be present in a ratio of 2.79:0.83:10.78:1.99:11.23:2.78, respectively [45]. Adzuki bean polysaccharides have great potential for biological activity, but there is not much research on them, and they are expected to have more research efforts.

### 2.5. Polyphenols

The study of polyphenols has been popular in bioactive substances of legumes in recent years. More than 8000 polyphenols have been identified in different plant species, where natural polyphenols mainly include phenolic acids, flavonoids, and tannins [46]. Phenolic substances in plants can be classified as soluble phenols and bound phenols, depending on if they are extracted directly or not by organic solvents. In addition to freeform, soluble phenolic extracts can also be bound in the form of ester, ether, and glycosidic bonds [47].

The content of polyphenols in adzuki beans may reach up to 3.73 mg/g, which is higher than those in both mung beans and soybeans [48], even reaching 10.38 mg/g, with 1.75 mg/g of free-state polyphenols and 8.63 mg/g of bound-state polyphenols. The content of bound phenols is about five times higher than the content of free phenols [49]. In Gan’s research, the total phenolic content extracted from adzuki beans was 13.37 mg/g [50]. In Sreerama’s study, the total phenolic content in adzuki beans (black) was 8.78 mg/g, and the total phenolic content in adzuki beans (red) was 4.89 mg/g [51]. The total phenolic contents of 17 Chinese adzuki bean varieties were 2.11~2.75 mg/g [8]. Table 1 shows different amounts of monomeric phenols, with 22 from Amarowicz, 6 from Yan, and 11 from Li, which determined the contents of monomeric phenols [52,53,54,55]. The differences in the polyphenol contents of adzuki beans may be due to the varieties and origins.

## 3. Processing Properties

Botanical starch has three main stereoscopic structure forms according to different XRD spectra patterns. Experiments have confirmed that type A starch is easier to digest than type B and C starches. Type A starch is common in cereal starch, type B starch is common in tubers, and type C starch is common in legumes and rhizomes. Studies have shown that azuki bean starch also belongs to C-type crystal structure starch [39,59]. Type A starch, with the components of short, lateral amylopectin chains and closed branching points, presents peaks at around 2θ angles of 15°, 17°, 18°, and 23°. Type B starch has long side amylopectin chains and distant branching points with clear diffraction peaks of 17°, while type C starch is a mixture of type A and type B, such that the special crystal structures consisting of type A and type B crystals give type C starch specific crystal tunability [60,61]. Azuki beans contain ideal, slow-digesting starch, and it is the semi-crystalline C-type structure of the starch that determines its slow-digesting properties. In common processing such as cooking, baking, and autoclaving, the destruction of crystal structure is disrupted, making the starch more digestible than unprocessed starch [62]. Significant correlations were also found between the starch content of azuki beans and the degree of pasting, pasting characteristics, hardness, and crystallinity [63]. Polysaccharides are another very important carbohydrate in legumes. The composition and molecular weight of polysaccharides play an important role in exerting bioactive effects [41]. A study confirmed some differences in composition between water-extracted and alkali-extracted polysaccharides from adzuki beans [42]. However, research on the processing and extraction of adzuki bean polysaccharides is very limited, and there are still many unsolved problems, which is a new research and exploration direction for scholars.

Compared to soybean protein, the adzuki bean protein isolate has high solubility at a lower pH. Under the same pH, the emulsification and oil-holding properties of adzuki bean protein remains basically the same as those of soybean protein isolate, but the water absorption, emulsification stability, and foaming stability are worse than those of soybean protein isolate. In addition, both adzuki bean and soybean proteins had fine gelation properties [64]. A study by Shen et al. confirmed that the solubility values of adzuki bean protein from five species reached the lowest values around pH 4.0. In addition, there was a correlation between the protein properties, including the surface hydrophobicity, emulsification, emulsion stability, water absorption, and oil absorption, of adzuki bean protein [65].

Yao et al. found that microwave baking and drum roasting influenced the generation of characteristic volatile compounds for adzuki beans, such as furan, pyrazine, ketone, alcohols, aldehydes, esters, pyrroles, sulfocompounds, phenols, and pyridine. Moreover, drum-roasted baked beans showed a higher flavor than that of microwave-baked beans. A critical temperature and low moisture content of 116.5 °C at a moisture content of 5.6% (w.b.) in microwave baking and 91.6 °C at a moisture content of 6.1% (w.b.) in drum roasting resulted in a rapid increase in acrylamide content [66].

Although adzuki beans are commonly consumed in daily life, their processing properties are still less studied. More research should be carried out on the processing characteristics of adzuki beans to develop a variety of adzuki bean series products, and it is of great significance to expand the consumption population of adzuki beans.

## 4. Healthy Benefits

Adzuki beans are also a traditional medicine that have been used as a diuretic and antidote, as well as to alleviate symptoms of dropsy and beriberi in China [3]. Therefore, research on adzuki beans and extracts of adzuki beans have long fascinated scientists.

Phytochemicals are non-nutritive, vegetative secondary components, with significant differences in biochemistry, source distribution, and physiological effects [67]. Their biological activities, such as antioxidant [68], anti-inflammatory [69], and antibacterial [70] properties, have also been described to provide important, meaningful benefits to human health. In studies on adzuki beans, it has been found that they are rich in phytochemicals, with important meaning for human health.

### 4.1. Antioxidant Activity

Legumes are rich in polyphenols, flavonoids, and proanthocyanidins, which are phytochemicals with natural antioxidants [71,72]. Already, a study in 2009 indicated that extracts from 15 common edible legumes had significant antioxidant activity [73]. The antioxidant properties of the total phenols and total flavonoids of five legumes, including black beans, mung beans, red adzuki beans, soybeans, and cowpeas, were investigated, and it was found that the scavenging abilities of 1,1-diphenyl-2-picrylhydrazyl free radicals (DPPH·), hydroxyl radicals (·OH) and superoxide radicals (O_2_^−^·) of the five legumes were relatively high [73]. Cowpeas, adzuki beans, mung beans, and broad beans had the ability of scavenging DPPH·, Ferric ion reducing antioxidant power (FRAP), 2,2′-azino-bis (3-e-htylbenzothiazoline-6-sulfonic acid) diammonium salt radical free radicals (ABTS+·), and O_2_^−^· [49].

For the antioxidant properties of adzuki beans, polysaccharides had great scavenging ability for O_2_^−^· and ·OH, which could effectively block the synthesis of nitrosamines, as well as having some scavenging ability for sodium nitrite [74]. Polyphenols extracted from adzuki bean coats were also shown to have scavenging abilities for DPPH·, O_2_^−^·, ABTS+·, ·OH, and the scavenging abilities for ABTS+· and ·OH were higher than that of vitamin C [50,57]. In addition to polyphenols and polysaccharides, condensed tannins showed a powerful antioxidant capacity for scavenging ABTS+· and FRAP [75]. Studies have confirmed the ability of components in adzuki beans to scavenge a variety of free radicals, which has led to the confirmation of the antioxidant capacity of adzuki beans.

### 4.2. Antimicrobial and Anti-Inflammatory Activity

Foodborne pathogenic bacteria contamination is a potential threat factor affecting food safety and triggering public health events. Common pathogenic bacteria include *Escherichia coli*, *Salmonella*, *Staphylococcus aureus*, and *Listeria monocytogenes*, etc. These pathogenic bacteria have strong resistance to environmental factors, such as dryness, acid, salt, and heat, resulting in their long-term survival during food production, processing, and storage [76].

It was shown in 2021 that the addition of adzuki bean coat polyphenol extract inhibited *Listeria* ATCC19119 and *Salmonella* ATCC14028 with minimum inhibitory concentrations of 625 μg/mL and 2500 μg/mL, respectively. The addition of adzuki bean coat polyphenol extracts had significant effects on the protein and nucleic acid contents, cell membrane potential, intracellular ATP content, extracellular alkaline phosphatase content, and morphological changes of the bacteria [56]. It was also demonstrated that the polyphenolic extracts in adzuki bean coats had inhibitory effects on two Gram-positive bacteria (*B. cereus* and *S. aureus*), but not on Gram-negative bacteria (*E. coli* and *S. typhimurium*) [50].

Inflammation is an over-reactive immune response of the body to infection or tissue damage that is associated with various diseases, such as inflammatory bowel disease, cancer, type 2 diabetes mellitus (T2DM), and cardiovascular disease [77,78]. In 2017, scientists confirmed the positive effects of black adzuki beans on colon inflammation triggered by high-fat-diet-induced obesity in mice. The black adzuki beans decreased the concentrations of lipopolysaccharides and circulating proinflammatory cytokines (such as tumor necrosis factor (TNF)- α, interleukin (IL)-1*β*, and IL-6) in mice [79]. Another study confirmed that an active peptide (KQS-1) isolated from extruded adzuki bean protein, sequenced as KQSESHFVDAQPEQQQR, exerted significant anti-inflammatory effects in lipopolysaccharide-induced RAW 264.7 macrophages and significantly reduced the production of IL-1, IL-6, TNF-α, and MCP-1 [80]. The antibacterial properties of adzuki beans reaffirm the potential function of extract from adzuki beans as a natural food preservative, and studies on the anti-inflammatory properties of adzuki beans also verify the medicinal and food properties of adzuki beans.

### 4.3. Antidiabetic Activity

It has been shown that the carbohydrates contained in legumes have a slow postprandial glycemic change and, therefore, most legumes are low-glycemic-index (GI) foods compared to starch-based food crops [81]. In a simulated in vitro digestion test of beans, it was found that the starch hydrolysis rate of adzuki beans was 31.42% after 2 h of in vitro digestion compared to 78% for rice, indicating a lower starch hydrolysis rate of adzuki beans compared to rice [82]. Therefore, adzuki beans have a great advantage in terms of starch digestive properties for an antidiabetic diet.

In 2005, scientists found that the hot water extract of adzuki beans had some inhibitory effects on α-amylase and glucosidase, which had some potential in the prevention of type II diabetes [83]. Research in 2012 also confirmed that adzuki beans had inhibitory effects on α-glucosidase and pancreatic lipase activities [51]. For further analysis of adzuki bean extract, researchers identified (+)-catechin 7-*O*-*β*-D-glucopyranoside (C7G), (+)-epicatechin 7-*O*-*β*-D-glucopyranoside (E7G), and (+)-catechin as the main bioactive components. These substances have an inhibitory effect on both *α*-amylase and α-glucosidase, and this inhibitory effect did not disappear after high-temperature heating [84].

Numerous experiments in mice have also confirmed the antidiabetic function of adzuki beans. Dietary supplementation with black adzuki bean extract significantly improved the hyperglycemia and homeostasis model assessment of insulin resistance index (HOMA-IR) in high-fat-diet-induced glucose-intolerant obese C57BL/6J mice [85]. Adzuki bean extract could potentially improve glucose intolerance in rats by upregulating the phosphorylation of serine/threonine protein kinase (AKT) and adenosine monophosphate-activated protein kinase (AMPK) in the livers of diabetic rats [86]. In addition, an oral administration of adzuki bean polysaccharides to rats could significantly reduce weight loss, fasting blood glucose (FBG), and the concentration of serum triglyceride (TG). In addition, it reversed dyslipidemia caused by diabetes, as evidenced by a reduction in triglycerides (TG) and elevated high-density lipoprotein cholesterol (HDL-C). With the increase in the expression of insulin receptor (INSR), insulin receptor substrate-1 (IRS-1), phosphoinositide 3-kinase (PI3K), protein kinase B (AKT), and glucose transporter-2 (GLUT-2) in type 2 diabetic rats, it indicated that adzuki bean polysaccharides regulated glucose metabolism by activating the PI3K/AKT signaling pathway, thus achieving an antidiabetic effect [87].

### 4.4. Hypolipidemic Activity

In addition to the antioxidant properties and its antidiabetic function, the hot water extract of adzuki beans was shown to inhibit lipid accumulation in 3T3-L1 adipocytes and to reduce body weight and adipose tissue weight [88]. Rats fed with adzuki bean paste were also found to reduce visceral fat accumulation and lower serum lipid levels in the adzuki-bean-paste-fed group more than the control group [89]. A high-fat diet feeding group supplemented with 15% adzuki beans and a control group were set up with mice and fed for 12 weeks. It was found that adzuki bean supplementation significantly reduced high-fat-diet-induced obesity, lipid accumulation, serum lipids, and lipopolysaccharide (LPS) levels, which mitigated liver function damage and hepatic steatosis [90].

When mice fed a high-cholesterol diet were given 0.5 mL of a solution of adzuki polyphenols for two weeks, the atherosclerotic index of experimental mice was significantly lower than that of the control group, indicating a significant inhibitory effect of adzuki polyphenols on serum cholesterol [91]. An 8-week trial evaluated in human subjects confirmed there was a significant increase in high-density liptein cholesterol concentrations in the participants receiving adzuki bean extract compared to a placebo, and no adverse effects were observed in the participants [92]. All of these studies have effectively confirmed the efficacy of adzuki beans in helping to resist to high-fat-diet-induced fat accumulation and in hypolipidemic activity, with no adverse effects in a safe manner.

### 4.5. Multiple Functions for Healthy Benefits

Adzuki beans have obvious medicinal and food homology. Diabetic kidney disease is associated with oxidative stress and inflammation. Scientists suggested that adzuki bean extract may weaken streptozotocin-induced diabetic kidney injury by inhibiting oxidation [93]. Another study also confirmed the positive effects of adzuki beans on the kidneys. Two surgeries on rats to obtain an animal model of moderate chronic kidney disease in the study of Baracho et al. revealed that the levels of glucose, triglycerides, very-low-density lipoproteins (VLDL), uric acid, alanine aminotransferase, urea, and serum creatinine were significantly reduced in mice in the adzuki-bean-treated group compared to the control and drug-treated groups, indicating the effective improvement of renal function parameters of adzuki bean [94].

Studies on the hot water extract of adzuki beans adsorbed onto DIAION HP-20 resin adsorption columns confirmed the inhibitory effect on experimental lung metastasis and the invasion of B16-BL6 melanoma cells, as well as the adhesion and migration of B16-BL6 melanoma cells to extracellular matrix components, and the study indicated that hot water extract of adzuki beans may have a strong antimetastatic ability [83]. A study showed an inhibitory effect of red bean extract (RBE) on muscle atrophy in an immobilized hindlimb muscle of C57BL/6J mice. Red bean extract increased the grip strength, exercise endurance, muscle weight, and muscle fiber area, which could significantly decrease the mRNA expression of proteolytic-related genes, such as muscle ring finger and muscle atrophy F-box, by the preventing the translocation of Forkhead box 3 [95]. Adzuki bean extract may reduce the elevation of blood pressure by regulating endothelial nitric oxide synthase (eNOS) and inducible nitric oxide synthase (iNOS) protein expression in the aorta and kidney [96]. High-fat-diet-induced obesity has been associated with cognitive and memory dysfunction. 

Studies have confirmed the contribution of adzuki beans to the treatment of obesity-induced cognitive decline. For high-fat-fed mice, adzuki bean extract could improve their spatial and recognition abilities. An increase in the ratios of their exploration of a novel object or new routes was also confirmed in a T-maze and novel object recognition tests [97]. A study on adzuki bean sprout fermented milk was also found to relieve anxiety and mild depression due to its richness in γ-aminobutyric acid (GABA) [98].

Adzuki beans have great health benefits, with a positive effect on the treatment of a number of diseases to some extent. The polysaccharides, polyphenols, and condensed tannins of adzuki beans have antioxidant effects; adzuki bean extract may reduce the elevation of blood pressure; adzuki bean coat polyphenol extract has antibacterial properties; and there is an inhibitory effect of adzuki bean extract on muscle atrophy. Additionally, adzuki bean extract and adzuki bean polysaccharides have been shown to have positive effects on high-fat-induced diabetes, while adzuki bean extract, polyphenols, and adzuki bean paste reduce high-fat-diet-induced obesity. Adzuki bean extract and adzuki bean sprout fermented milk play a role in the remission of obesity-induced cognitive decline and the relief of anxiety and mild depression. Adzuki beans also could mitigate liver function damage and hepatic steatosis, while adzuki beans and their extracts have positive effects on the kidneys and diabetic kidney disease (Figure 2).

Adzuki beans have many health benefits. Because of its slow digestibility, adzuki bean starch can slowly increase the glycemic index, which has a positive effect on the prevention and treatment of diabetes. Many studies have confirmed that polysaccharides are the main active components of legumes and have a variety of biological activities, including antioxidant and immunomodulatory activities [41,42,43]. Adzuki beans and their extracts have positive effects on the treatment of many diseases, which may also be because of their polysaccharide components. In addition, the polyphenols of adzuki beans also play a very important role in many aspects, such as antioxidant and anti-inflammatory activity.

## 5. Conclusions and Future Perspective

As a traditional Asian food, there are many varieties of adzuki beans with differences in starch, protein, and fat contents and compositions that are rich in polysaccharides and polyphenols. Due to the starchy nature of the beans and the phytochemicals, the consumption of adzuki beans has various health benefits, which display positive effects on the treatment of diseases. This establishes the feasible edibility of adzuki beans and their medicinal value.

The domestication and research on adzuki beans have been conducted for many years, but there are still some problems in the process of research. Bioactive components such as phenols and polysaccharides in adzuki beans have been confirmed to play an important role in human health, but there are no relevant standards for the extraction of these substances, resulting in differences in the identification and contents of substances due to differences in extraction methods in the research process. This also provides an obstacle for the further study of adzuki bean bioactive substances. Adzuki bean extract has a positive effect on many diseases, but the components of adzuki bean extract are complex, and the identification and study of effective monomer components are limited. In addition, studies on the polyphenols and polysaccharides of adzuki beans are often conducted by in vitro experiments through extracts. However, the utilization rate of polyphenols and polysaccharides in adzuki beans after eating, as well as how to maximize the retention of the bioactive substances of adzuki beans in processing, is still little-studied. The change in the structure of polysaccharides may lead to a change in their biological activity, and chemical, physical, and biological methods can cause the change in the structure of polysaccharides [99]. However, the changes in polysaccharides and phytochemicals during processing and the bioactivity of adzuki beans after processing need to be further studied. Edible legumes have some potential for intestinal probiotics, but there are few studies related to adzuki beans. To date, less research has been performed on adzuki beans at the genetic level. In future studies, it is expected that more research can participate in the study of adzuki beans, and the processed food of adzuki beans can be enjoyed by more people.

## Figures and Tables

**Figure 1 molecules-27-06079-f001:**
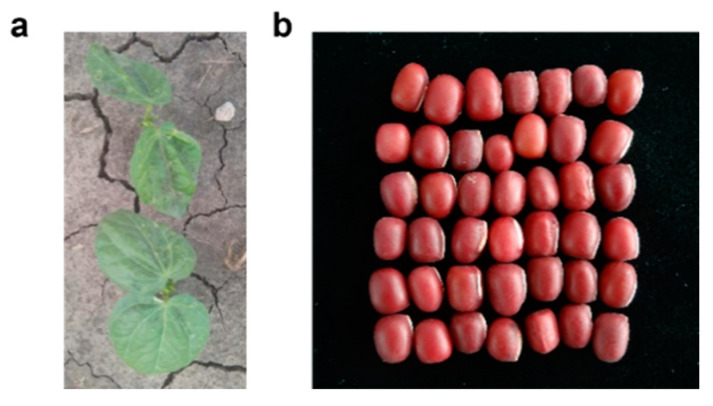
Adzuki bean. (**a**) Seedlings of adzuki beans (*Vigna angularis*) in the field [7]. (**b**) Seeds of adzuki beans (*Vigna angularis*).

**Figure 2 molecules-27-06079-f002:**
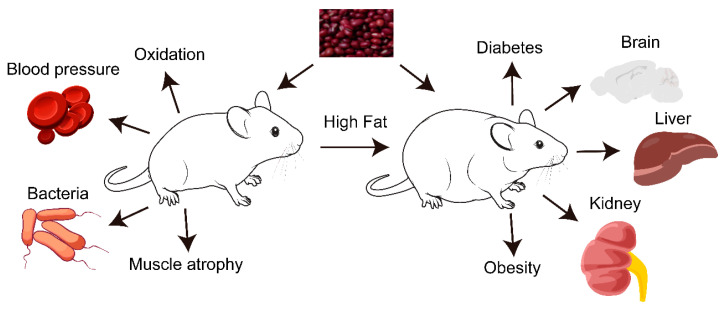
Health benefits of adzuki beans and adzuki bean extract in mice or rats.

**Table 1 molecules-27-06079-t001:** Phenolic compounds and their contents in adzuki beans.

Location of Extraction	Name of Compound	Analytical Method	Content (μg/g)
Adzuki Bean	Protocatechuic acid [52,55]	UPLC-QTOF-MS and HPLC-QQQ-MS [52], HPLC-PAD [55]	67.6 ± 4.01 [55], 13.94 ± 0.22 [52]
Adzuki Bean	Protocatechuic aldehyde [55]	HPLC-PAD [55]	7.71 ± 0.62 [55]
Adzuki Bean/Adzuki Bean Coat	*trans*-*p*-Coumaric acid [52,55,56]	UPLC-QTOF-MS and HPLC-QQQ-MS [52], HPLC-PAD [55], HPLC [56]	31.3 ± 1.96 [55], 7.16 ± 0.06 [52]
Adzuki Bean	*trans*-*p*-Coumaroyl malic acid [55]	HPLC-PAD [55]	4.57 ± 0.69 [55]
Adzuki Bean	Epicatechin [55]	HPLC and LC/MS [54], HPLC-PAD [55]	25.7 ± 2.06 [55]
Adzuki Bean	Epigallocatechin gallate [55]	HPLC-PAD [55]	0.14 ± 0.02 [55]
Adzuki Bean	Epicatechin glucoside [55]	HPLC-PAD [55]	159.0 ± 8.31 [55]
Adzuki Bean	Catechin glucoside [55]	HPLC-PAD [55]	688.0 ± 35.6 [55]
Adzuki Bean/Adzuki Bean Coat	Quercetin [52,54,55,56,57]	UPLC-QTOF-MS and HPLC-QQQ-MS [52], HPLC and LC/MS [54], HPLC-PAD [55], HPLC [56,57]	36.2 ± 1.54 [55], 3.07 ± 0.05 [52]
Adzuki Bean	Quercetin rutinoside [55]	HPLC-PAD [55]	38.2 ± 1.52 [55]
Adzuki Bean	Quercetin galactoside [55]	HPLC-PAD [55]	46.9 ± 5.87 [55]
Adzuki Bean	Quercetin glucoside [55]	HPLC-PAD [55]	181.0 ± 9.14 [55]
Adzuki Bean	Quercetin arabinoglucoside [55]	HPLC-PAD [55]	42.8 ± 4.23 [55]
Adzuki Bean	Dihydroquercetin [55]	HPLC-PAD [55]	1.15 ± 0.07 [55]
Adzuki Bean	Dihydroquercetin hexose [55]	HPLC-PAD [55]	0.54 ± 0.02 [55]
Adzuki Bean	Dihydroquercetin derivative [55]	HPLC-PAD [55]	1.35 ± 0.06 [55]
Adzuki Bean	Myricetin rhamnoside [55]	HPLC-PAD [55]	212.0 ± 9.85 [55]
Adzuki Bean	Kaempferol rutinoside [55]	HPLC-PAD [55]	38.2 ± 1.52 [55]
Adzuki Bean	Tetrahydroxydihydrochalcone glycoside [55]	HPLC-PAD [55]	0.55 ± 0.08 [55]
Adzuki Bean	Procyanidin gallate [55]	HPLC-PAD [55]	12.4 ± 1.06 [55]
Adzuki Bean	Procyanidin dimer [55]	HPLC-PAD [55]	213.0 ± 13.2 [55]
Adzuki Bean	Procyanidin trimer [55]	HPLC-PAD [55]	41.8 ± 1.11 [55]
Adzuki Bean	Kaempferol-3-*O*-*β*-*D*-glucoside [53]	HPLC-ESI-TOF-MS [53]	1.6 ± 0.1 [53]
Adzuki Bean	Isoquercitrin [53]	HPLC-ESI-TOF-MS [53]	0.8 ± 0.03 [53]
Adzuki Bean/Adzuki Bean Coat	Rutin [50,52,53,54,56,57]	UPLC-QTOF-MS and HPLC-QQQ-MS [52], HPLC-ESI-TOF-MS [53], HPLC and LC/MS [54], HPLC-PAD [55], HPLC [56,57]	420 ± 0.02 [50], 4.1 ± 0.2 [53], 327.40 ± 5.43 [52]
Adzuki Bean	Isovitexin-6′-*O*-*α*-*L*-glucoside [53]	HPLC-ESI-TOF-MS [53]	1.3 ± 0.03 [53]
Adzuki Bean/Adzuki Bean Coat	Chlorogenic acid [52,53,56,57]	UPLC-QTOF-MS and HPLC-QQQ-MS [52], HPLC-ESI-TOF-MS [53], HPLC-PAD [55], HPLC [56,57]	14.0 ± 1.3 [53], 1.76 ± 0.05 [52]
Adzuki Bean	Benzyl-*O*-*β*-*D*-glucopyranoside [53]	HPLC-ESI-TOF-MS [53]	0.4 ± 0.01 [53]
Adzuki Bean Coat	Gallic acid [56,57]	HPLC [56,57],	—
Adzuki Bean/Adzuki Bean Coat	Catechin [52,54,56,57]	UPLC-QTOF-MS and HPLC-QQQ-MS [52], HPLC and LC/MS [54], HPLC [56,57]	210.70 ± 5.58 [52]
Adzuki Bean Coat	Epicatechin [56,57]	HPLC [56,57]	—
Adzuki Bean Coat	Procyanidin B_2_ [56,57]	HPLC [56,57]	—
Adzuki Bean Coat	Ferulic acid [56,57]	HPLC [56,57]	—
Adzuki Bean Coat	Isovitexin [57]	HPLC [57]	—
Adzuki Bean Coat	Vitexin [57]	HPLC [57]	—
Adzuki Bean Coat	Isorhamnetin [57]	HPLC [57]	—
Adzuki Bean/Adzuki Bean Coat	Hyperoside [52,56,57]	UPLC-QTOF-MS and HPLC-QQQ-MS [52], HPLC [56,57]	2.59 ± 0.05 [52]
Adzuki Bean Coat	Kaempferol [56,57]	HPLC [56,57]	—
Adzuki Bean Coat	Syringic acid [56]	HPLC [56]	—
Adzuki Bean Coat	Caffeic acid [56]	HPLC [56]	—
Adzuki Bean	4-Hydroxybenzoic acid [52]	UPLC-QTOF-MS and HPLC-QQQ-MS [52]	0.52 ± 0.12 [52]
Adzuki Bean	Luteolin [52]	UPLC-QTOF-MS and HPLC-QQQ-MS [52]	0.16 ± 0.01 [52]
Adzuki Bean	Daidzein [52]	UPLC-QTOF-MS and HPLC-QQQ-MS [52]	3.11 ± 0.19 [52]
Adzuki Bean	Glycitein [52]	UPLC-QTOF-MS and HPLC-QQQ-MS [52]	0.09 ± 0.01 [52]
Black Adzuki Bean	Delphinidin-3,5-*O*-digalactoside [58]	NMR and UPLC-Q-Orbitrap-MS/MS [58]	—
Black Adzuki Bean	Delphinidin-3,5-*O*-diglucoside [58]	NMR and UPLC-Q-Orbitrap-MS/MS [58]	—
Black Adzuki Bean	Delphinidin-3-*O*-galactoside [58]	NMR and UPLC-Q-Orbitrap-MS/MS [58]	—
Black Adzuki Bean	Delphinidin-3-*O*-glucoside [58]	NMR and UPLC-Q-Orbitrap-MS/MS [58]	—
Black Adzuki Bean	Delphinidin-3-*O*-rutinoside [58]	NMR and UPLC-Q-Orbitrap-MS/MS [58]	—
Black Adzuki Bean	Delphinidin-3-*O*-(*p*-coumaroyl) glucoside [58]	NMR and UPLC-Q-Orbitrap-MS/MS [58]	—
Black Adzuki Bean	Cyanidin-3-*O*-glucoside [58]	NMR and UPLC-Q-Orbitrap-MS/MS [58]	—
Black Adzuki Bean	Petunidin-3-*O*-galactoside [58]	NMR and UPLC-Q-Orbitrap-MS/MS [58]	—
Black Adzuki Bean	Petunidin-3-*O*-glucoside [58]	NMR and UPLC-Q-Orbitrap-MS/MS [58]	—
Black Adzuki Bean	Petunidin-3-*O*-(*p*-coumaroyl) glucoside [58]	NMR and UPLC-Q-Orbitrap-MS/MS [58]	—

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
