# Peer review of "Nutritional Composition, Efficacy, and Processing of *Vigna angularis* (Adzuki Bean) for the Human Diet: An Overview"

_molecules, 2022, doi:10.3390/molecules27186079_

Round 1

Reviewer 1 Report

The current manuscript based in the review of the  nutritional composition and health benefits associated to Adzuki bean shows several drawbacks which must be attended before its publication in Molecules journal.

Point 1. It should be important to mention the relevance of Adzuki bean for European and American regions since introductory section is mainly focused for the Asian community

Point 2. Line 35-37. What is the relevance of the number of chromosomes and genome mapped?

Point 3. In 2.4 polysaccharides section, it is recommendable to mention what type of polysaccharides are found in Adzuki beans. Please give more details about it.

Point 4. Regarding processing properties, the C-type structure of starch must be described. Further, the structure of other polysaccharides must be included in the description as well as the technological properties related to these carbohydrates.

Point 5. The health benefits must be mainly attributed to carbohydrates (starch and other polysaccharides) instead other type of bioactive compounds, such as polyphenols. Please revise it.

Point 6. Conclusion must be improved mainly highlighting the possible changes of functional or health properties as consequence of possible alteration of its polysaccharides and phytochemicals by processing. 

Reviewer 2 Report

The review investigated the evaluation of the nutritional composition, efficacy, and processing of Vigna angularis (Adzuki Bean) in the human diet. The review covered nutritional composition, processing properties, as well as health benefits. The review is very promising, and can be further improved based on the following:
a) Abstract needs improvement, it does not clearly identify the objective of the review, and why this review is warranted, prior to expanding on key aspects of the review.
b) Introduction should clearly expand on beans more in general, challenges the crop encountered across the globe, prior to narrowing it down to vigna sp.. When presenting vigna, do not restrict it to China, give it a global perspective, before narrowing it down to China. Then, in China, identify why Adzuki beans is largely favored. All these should come before Line 40.
Expand Lines 40-57, to include, why is this review relevant? Identify with previous reviews conducted on Adzuki beans, if non-exists, demonstrate it clearly. Reviewer suggests authors should create a table, identifying with previous recently conducted reviews, within the past 5- 10 years on vigna sp., the table should have, columns should include a) objective of review,b) key aspects captured, and c) references. 
The reviewer will look out for this in the revised manuscript
c) The subsequently sections of the review, before conclusions/future perspective is ok.
d) Please, combine 'Conclusions and Future Perspectives" as one section/paragraph.
Overall, given the succinct nature of this review, it is better to title reads: "Nutritional composition, efficacy, and processing of Vigna angularis (Adzuki Bean) for the human diet. An overview"
Look forward to your revised manuscript.
d) body of the work appears

Round 2

Reviewer 1 Report

The authors attended all points in a good way. 

Reviewer 2 Report

The authors have done a brilliant effort to revise their work.

It is now acceptable for publication. Thank you very much